# Comparison of thoracic ultrasonography (TUS), clinical respiratory scoring (CRS), and blood analysis to evaluate respiratory dysfunction in transported calves

Luca L. van Dijk[1,2*], Susanne Siegmann[1,3], Niamh L. Field[1], Katie Sugrue[1], Cornelis G. van Reenen[3], Eddie A.M. Bokkers[3], Gearoid Sayers[2], Muireann Conneely[1]

**1** Teagasc, Animal and Grassland Research and Innovation Centre, Moorepark, Fermoy, Ireland, **2** Department of Biological and Pharmaceutical Sciences, Munster Technological University Kerry, Tralee, Ireland, **3** Animal Production Systems group, Wageningen University and Research, Wageningen, the Netherlands

* Luca.vanDijk@anivet.au.dk (LD); Muireann.Conneely@Teagasc.ie (MC)

## Abstract

Mixing of calves during and after transport exposes them to large numbers of pathogens, increasing their risk of developing bovine respiratory disease after arrival. Identifying individual calves with respiratory disease is important to assess transport fitness and to facilitate timely treatment. This study compared three screening tests (thoracic ultrasound scores [TUS], clinical respiratory scores [CRS], and blood variables indicative of immune and inflammation status) for detecting respiratory disease in calves before and after transport to assess the degree of agreement in identifying respiratory dysfunction in transported calves. Thoracic ultrasound scores (0: healthy to 5: severe pneumonia; TUS of 4 and 5 were rare and reclassified as TUS 3 for analysis), CRS (0: healthy, 1: mild, 2: moderate, 3: severe), and blood variables (haemoglobin; red and white blood cell, neutrophil, lymphocyte, monocyte, basophil, and eosinophil counts; and neutrophil:lymphocyte ratio; serum-amyloid-A; total immunoglobulins; and immunoglobulins –A, -G, and –M) were compared in 65 calves at five time points; pre-transport (T1), on arrival (T2), 3 or 7 days post-arrival (T3; TUS and CRS only), and approximately ten (T4) and eighteen days (T5) post-arrival. Comparatively, TUS and CRS had different distributions between time points with TUS gradually increasing between T1 and T3 and declining thereafter, while CRS rapidly increased between T1 and T2, followed by a gradual decrease. Lymphocyte count was lower for a TUS of 3 than for a TUS of 1 or 2, serum-amyloid-A was higher for a TUS of 2 or 3 than for a TUS of 0, total immunoglobulins and immunoglobulin-G were lower for TUS of 3 than for a TUS of 2, and immunoglobulin-M was higher for a TUS of 1 or 2 than for a TUS of 0. Haemoglobin, WBC, and lymphocyte count were lower for CRS of 1 or 2 than for a CRS of 0. In conclusion, there was no agreement between TUS and CRS on the severities of individual scores. Though lymphocyte

**Data availability statement:** All relevant data are within the paper and its Supporting Information files.

**Funding:** The author(s) received no specific funding for this work.

**Competing interests:** The authors have declared that no competing interests exist.

count was lower for moderate or severe CRS or TUS, no other blood variables showed a consistent pattern of change in relation to TUS or CRS.

## Introduction

Clinical respiratory scoring (CRS) is routinely used by exporters and veterinarians as one of the indicators to determine a calf's fitness for transport. Bovine respiratory disease is the main disease of concern, and transport-associated respiratory disease (or "shipping fever") is deemed a main cause of morbidity in transported calves, as pathogen loads are higher during and post transportation [1]. Respiratory disease is caused by viral or bacterial pathogens and characterised by inflammation of the upper and/or lower respiratory tract [2] and is responsible for major economic losses in dairy and veal systems, particularly due to treatment costs, decreased growth, and mortality [3]. Calves with clinical signs pre-transport may infect other calves and may be at higher risk of severe physiological decline and impaired welfare during transport [4]. Correctly identifying calves with subclinical or clinical (respiratory) disease prior to transport will reduce disease prevalence post transport, thereby improving calf welfare, reducing costs for the farmer, and reducing antibiotic use [5].

Clinical respiratory scoring is based on a visual assessment of eye discharge, nose discharge, ear position, cough frequency and rectal temperature [6]. Even though CRS is routinely used to assess calves prior to transport, its low sensitivity (between 0.27 and 0.64, with a specificity estimated from 0.74 to 0.94; measured on-farm) [5] renders it an unreliable screening tool to detect calves unfit for transport [6]. Several other screening tests exist for the identification of respiratory disease in calves, including thoracic auscultation (AUS), thoracic radiography (TR), thoracic ultrasonography (TUS), and measurements of various blood variables. In terms of accuracy, AUS has wider reported ranges than CRS, with sensitivities ranging from 0.06 to 0.90 and specificities ranging from 0.46 to 0.99 [6]. Thoracic radiography is likely the most sensitive test (sensitivity: 0.86, specificity: 0.89; [7]), but does not fit the criteria as a screening test, because it requires highly trained radiographers, which is expensive and impractical on farms [8]. Thoracic ultrasonography, though labour-intensive, offers the ability to detect subclinical respiratory disease and assess the severity of clinical BRD [9]. It has sensitivities of over 0.85 (up to 0.94), while specificities range from 0.98 to 1.0 when the severity of TUS is compared to necropsies [10,11]. Various blood variables change during respiratory disease, neutrophilia, monocytosis, lymphopenia, and increased SAA have been observed in response to respiratory infections in calves [9,12–17]. Blood variables take time to analyse, which means they don't typically qualify as a screening test. However, in veal systems, blood samples are already collected routinely. This means immune-related blood markers could be a useful extra tool for detecting respiratory disease, especially when used alongside other methods.

In the context of live transport, the ability to differentiate calves that are fit for transport from those that are not, represents a primary welfare standard that must be

accurate in order to be effective. Therefore, the aim of this study was to compare three screening tests (TUS, CRS, and blood variables indicative of immunity and inflammation) for the assessment of respiratory dysfunction, potentially respiratory disease, in calves before and after long distance transport.

## Materials and methods

### Calf selection

Calves were selected during two consecutive commercial transport cohorts from Ireland to the Netherlands in April (cohort 1; C1) and May (cohort 2; C2) of 2022. Eight dairy farms (Munster, Ireland) were selected by a collaborating exporter based on at least two calves available for the set departure date from each farm. All dairy farms were visited by the project team and all calves over 14 days of age that were presented for export were enrolled in the trial. Additionally, one commercial livestock mart (sale yard) for each cohort (C1 and C2) in Munster, Ireland, were visited and 20 calves per cohort were bought at the commercial mart and enrolled in the trial. C1 consisted of 37 calves, all male (5 HF and 32 BeefXHF), average age 29.6d (16 – 42d), average weight 55.1 kg (44–67 kg) at the assembly centre. C2 consisted of 29 calves, 20 males (10 HF and 10 BeefXHF) and 9 females (all BeefXHF), average age 28.6d (17 – 36d) and average weight 57.3 kg (43–72 kg) at the assembly centre. The cohorts of calves used in this study were part of a larger trial examining calves during long distance transport from Ireland.

### Transport and feeding

The approximate transport timeline between the farm/mart of origin in Ireland to the veal farm in The Netherlands is presented in Fig 1. This study followed an observational format; calves were transported commercially and standard transport, feeding, and housing procedures were not altered for this study. Calves originating from dairy farms were assessed at the dairy farm of origin one day before departure to the assembly centre, while calves that entered the trial at the livestock mart were assessed at the livestock mart on the day of transport to the assembly centre. From the dairy farm or livestock mart, calves were transported to a commercial export centre (Leinster, Ireland) where they arrived around midnight. They were offloaded into straw-bedded pens and rested overnight. The following morning, calves were fed 2 L of milk replacer [21% protein, 17% fat] and loaded onto a commercial three-tier livestock lorry in the afternoon. The lorry travelled to Dublin port (C1) or Rosslare port (C2) where it boarded a ferry to Cherbourg, France. On arrival in France, the lorry drove to a control post (Normandy, France), where calves were unloaded, fed 3 L of milk replacer [22% protein, 19% fat], and rested. After 13 hours at the lairage, calves were re-loaded onto the lorry and transported to a commercial veal farm in the Gelderland province of The Netherlands. On arrival at the veal farm, calves were unloaded and placed randomly in individual pens (approximately 1 m x 1.6 m) on wooden slats. Calves were separated by stainless steel fencing but were able to see and touch calves in neighbouring pens. All housing complied with European legislation for veal calves. Calves remained individually housed for 3 weeks post arrival. Calves were fed according to standard veal farm protocols; they were fed an electrolyte mix in 2 L of water for the first feed post arrival, from the next feeding onwards calves were fed 1.5 L of milk replacer [21% protein, 18.2% fat] twice a day in buckets, the volume gradually increased to 2.7 L twice a day by 3 weeks post arrival.

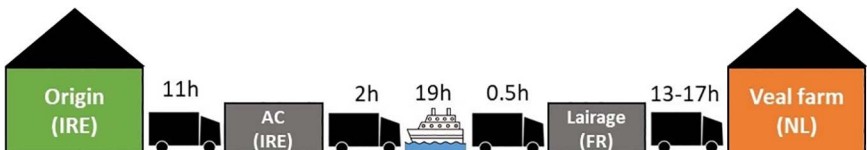

**Fig 1. Timeline of transport, including farm/mart of origin (Origin), assembly centre (AC), lairage (Lairage), and destination veal farm (Veal farm), and approximate transport times between resting destinations.** Locations were in Ireland (IRE), France (FR), or The Netherlands (NL).

All calves received antibiotic treatment shortly after arrival (Tilmicosin: 1 cc I.M.), and received an additional two courses of metaphylactic batch antibiotics in week 1 and week 2 post arrival at the veal farm (Doxycycline: 0.03 g per feed in milk twice a day for three and five days respectively). No animals were euthanized for this study. Analgesia or euthanasia was provided only at the discretion of the farmer and attending veterinarian, and all handling was performed to minimize stress.

### TUS and CRS scoring

Table 1 presents the sampling schedule for TUS, CRS, and blood variables for calves transported in C1 or C2 in this study. Calves were scored (CRS and TUS) once pre-transport at the dairy farm/mart of origin (T1). On the destination farm calves were scored (CRS and TUS) one or two days post arrival (T2), three or seven days post arrival (T3), nine or ten days post arrival (T4), and 18 days post arrival (T5). The same observer assessed CRS and TUS of calves at all time points. Calves were blood sampled once pre-transport (dairy farm/mart; T1), on arrival at the veal farm (T2), and 10 days (T4) and 21 days post arrival (T5) at the veal farm. No blood samples were collected between arrival and 10 days post arrival (T3 time point) due to logistical constraints.

TUS was performed by a trained observer based on a system used by Ollivett and Buczinski [18] which was slightly adapted for the current study. As a preparation for scanning, the hair of the calf in the lung area was drenched in 70% isopropyl alcohol to allow for ultrasound waves to be undisturbed by air. The ultrasound probe was moved ventrally in each intercostal space, the pattern was repeated in a cranial direction from the 12th to the 3rd (left) or to the 1st (right) intercostal space using a 4.5–8.5 MHz ultrasound scanner (Easi-Scan veterinary ultrasound scanner, IMV Imaging; Gormanston, Ireland) set to a depth of 11 cm using a lower frequency to obtain a greater penetration (setting: late-term). The right front leg of the calf was lifted to reach the 2nd and 1st intercostal space on the right. An overall 6-tier TUS was based on the score of four lobes (right cranial, right caudal, left cranial, and left caudal). The observer placed the ultrasound probe dorsally near the spine in the 12th intercostal space and moved ventrally until visualisation showed the end of the lung tissue within a single intercostal space, after which this step was repeated for the next cranially adjoining intercostal space. The TUS system used is presented in Table 2. A 6-point TUS scale was used for observations on farm, but due to low observations of TUS 4 and 5, these were reclassified as TUS of 3. Thoracic ultrasound scores of 0 were considered healthy, TUS of 1 were considered to have mild respiratory disease signs, TUS of 2 were considered to have moderate respiratory disease signs, and TUS ≥ 3 were considered to have severe respiratory disease signs.

Clinical respiratory scoring was conducted shortly before or after the TUS observations for every time point. The CRS system was based on the Wisconsin Calf Health Scoring Chart [19] and adapted for practical use in transport research (Table 3). Rectal temperatures were not taken at T1 due to practical limitations and were omitted for the CRS calculation at this time point. Furthermore, for 36 calves, rectal temperatures from 9 days post arrival were used for CRS calculations day 10 post arrival due to missing temperatures for these calves on this day. Individual clinical respiratory scores are presented in Table 3. The total sum of clinical respiratory scores was calculated as the sum of all individual clinical respiratory scores included in Table 3 (total score = rectal temperature score + coughing score + nasal discharge score + eye discharge

**Table 1. Sample schedule for blood sampling (Blood), thoracic ultrasonography (TUS) and clinical respiratory (CRS) scoring on days relative to arrival (Day 0). Calves (n = 66) departed their dairy farms in Ireland on Day −3 and arrived at the veal farm in The Netherlands on Day 0.**

| Time point | T1 | | | | T2 | | | T3 | | | | | | | T4 | | | | T5 | | |
|---|---|---|---|---|---|---|---|---|---|---|---|---|---|---|---|---|---|---|---|---|---|
| **Cohort 1** | | | | | | | | | | | | | | | | | | | | | |
| Blood (n) | 17 | 20 | | | 37 | | | | | | | | | | | | 37 | | | | 37 |
| TUS/CRS (n) | 17 | 20 | | | | 37 | | 37 | | | | | | | 10 | 27 | | | 37 | | |
| **Cohort 2** | | | | | | | | | | | | | | | | | | | | | |
| Blood (n) | 10 | 19 | | | 29 | | | | | | | | | | | | 29 | | | | 29 |
| TUS/CRS (n) | 10 | 19 | | | | | 29 | | | | | 29 | | | | 29 | | | 29 | | |
| **Day** | −4 | −3 | −2 | −1 | 0 | 1 | 2 | 3 | 4 | 5 | 6 | 7 | 8 | 9 | 10 | 11 | 12 - 17 | 18 | 19 | 20 | |

**Table 2. Thoracic ultrasound score system[1].**

| TUS | Description |
|---|---|
| 0 | Normal aerated lung in the four examined lobes; indicating healthy lung tissue |
| 1 | Diffuse comet-tail artefacts in one to four lobes; indicating localised mild inflammation |
| 2 | Lobular or patchy consolidation in one or more lobes, yielding a theoretical total equivalence loss of functioning of less than one whole lobe; indicating moderate inflammation likely with minimal loss of functioning |
| 3 | Lobar consolidation in one lobe, or multiple lobes presenting with lobular consolidation yielding a theoretical total equivalence of loss of functioning of one whole lobe; indicating severe inflammation likely with tissue damage and reduced function in one lobe |
| 4 | Lobar pneumonia in two lobes; indicating severe inflammation and likely loss of functioning in two lobes |
| 5 | Lobar pneumonia in three or more lobes; indicating severe inflammation and likely loss of functioning in three or more lobes |

[1]Adapted from a scoring system used by Ollivett et al. [18]

**Table 3. Clinical respiratory variables and point scale used for health scoring of calves undergoing transportation from Ireland to The Netherlands[1].**

| Clinical parameter | Score and description | | | |
|---|---|---|---|---|
| | 0 | 1 | 2 | 3 |
| Rectal temperature (°C) | < 38.3 | 38.3 - 38.7 | 38.8- 39.4 | > 39.4 |
| Coughing | No cough | Single cough | Repeated cough (within ± 2 minutes) | – |
| Nasal discharge | Normal, serous discharge | Small amount of unilateral, cloudy discharge | Bilateral, cloudy, or excessive mucus | Copious, bilateral |
| Eye discharge | No eye discharge | Small amount of unilateral discharge (<1 cm2) | Moderate amount of bilateral discharge | Heavy ocular discharge |
| Ear position | Normal ear position | Head shaking | Slight unilateral ear droop | Bilateral ear droop |

Total scores were calculated as:

rectal temperature + coughing + nasal discharge + eye discharge + ear position.

CRS were combined to fit four categories:

total score 0 or 1 = CRS of 0, total score 2 or 3 = CRS of 1, total score 4 or 5 = CRS of 2 and total score ≥ 6 = CRS of 3

[1]Adapted from scoring systems used by McGuirk and Peek [19].

score + ear position score). To equally compare CRS and TUS, an equal number of categories was favoured, therefore, total CRS scores were confined to four categories, a total score of 0 or 1 = CRS of 0, a total score of 2 or 3 = CRS of 1, a total score of 4 or 5 = CRS of 2 and a total score ≥ 6 = CRS of 3. Clinical respiratory scores of 0 were considered to be healthy, CRS of 1 were considered to have mild respiratory disease signs, CRS of 2 were considered to have moderate respiratory disease signs and CRS of 3 were considered to have severe respiratory disease signs.

## Blood sampling and laboratory processing

For every blood sample, 26 ml was taken by jugular venepuncture using a 20 G needle by experienced personnel. This corresponds to 0.63% of blood volume for a 55 kg calf and 0.81% for the lightest calf (43 kg), assuming total blood volume is 75 mL/kg for a calf between 16 and 30 days [20], both below the 1% safety threshold for sampling of total blood volume within 24 h [21]. Blood was collected into four different tubes containing EDTA (6 ml), heparin (6 ml), glycolytic inhibitor (6 ml), or serum separator (SST; 8.5 ml). EDTA tubes were refrigerated, and haematology (haemoglobin (HGB), white blood cell (WBC), red blood cell (RBC), neutrophil, lymphocyte, monocyte, basophil, and eosinophil counts) were analysed. Samples collected in Ireland (SF/MA) were analysed using the Advia 2120 system (Bayer, AG) at Teagasc Grange (Dunsany, Ireland) while all other samples were analysed using fluorescence flow cytometry (XT-1800i, Sysmex Europe GmbH, Germany) at Rimondia

(Elspeet, the Netherlands). Haematology was analysed in different locations to minimise time between sample collection and laboratory analysis of fresh samples. All haematology samples were analysed within 48h of collection. Assay performance was ensured through the inclusion of quality control samples, for Teagasc Grange samples %CV was calculated in low, normal, and high quality control samples for HGB (low: 1.31%, normal: 0.89% and high: 1.73%) and neutrophil count (low: 4.75%, normal: 2.33%, and high: 3.21%). For Rimondia samples, only normal quality control samples were included, which yielded a %CV of 0.78% for HGB and 3.20% for neutrophil count. Haemoglobin was analysed in different units in Ireland (g/dL) and The Netherlands (mmol/L), those analysed in g/dL were transformed to mmol/L (multiplied by 0.6206; [22]).

All remaining samples were spun at 3,000 rpm for 5 min (heparin and glycolytic inhibitors) or 10 min (SST), decanted into serum tubes, and stored in a freezer (−20°C). All serum tubes were sent to Teagasc Grange (Dunsany, Ireland), ELISA was used to analyse serum amyloid-A (SAA) (Tri Delta TP802, Maynooth, Ireland), immunoglobulin-A (IgA) and Immunoglobulin-G (IgG) (Eagle BGG69-KOI, Amherst, NH, USA), and immunoglobulin-M (IgM) (Eagle BCM61-KOI, Amherst, NH, USA).

## Statistical analysis

Prior to statistical analysis, one BeefXHF calf transported in C1 was removed from the analysis due to a missing TUS observation. In total, 65 calves were included in the analysis. A TUS of 5 was not registered in this study and a TUS of 4 was recorded in only four cases (out of 325 observations), which, for statistical purposes, were reclassified as a TUS of 3. For data presenting with extreme outliers, values were removed when they were greater than the mean±3*SD. As a result, two IgM values, two neutrophil/lymphocyte ratio values, and four eosinophil values were removed. At T3, no blood samples were collected, therefore TUS and CRS comparisons to blood samples were confined to T1, T2, T4, and T5 time points.

All statistical analyses were performed using SAS on Demand (SAS Institute; NC, USA). For the descriptive statistics, changes between time points were calculated for TUS and CRS (i.e., whether the TUS value at T2 was higher, equal, or lower than the TUS value at T1), and presented as a heat map. To describe changes in blood variables across time points, a minimal model was used to analyse changes in blood variables over time. The model included a fixed effect of Time (T1, T2, T4, and T5) using a repeated effect of time with calf as the subject. It was re-run for every dependent variable (blood variables; RBC, WBC, neutrophil, lymphocyte, monocyte, basophil, and eosinophil counts, and HGB, SAA, total immunoglobulins (Total IG; sum of IgA, IgG, and IgM), and IgA, IgG, and IgM). Variables for which residuals failed normality tests upon initial analysis (all except for RBC, HGB, and lymphocyte count) were analysed using a lognormal distribution.

For the comparison of TUS and CRS to blood variables, the model included a fixed effect (TUS=0, 1, 2, 3, or CRS=0, 1, 2, 3), and a random effect of calf, and was re-run for every blood variable (RBC, WBC, neutrophil, lymphocyte, monocyte, basophil, and eosinophil counts, and HGB, SAA, Total IG, IgA, IgG, and IgM). For TUS versus blood variables, RBC was analysed using a normal distribution, lymphocyte count and SAA were analysed as square root-transformed variables, while the remaining variables were analysed using a lognormal distribution. For CRS versus blood variables, RBC, and lymphocyte count were analysed using a normal distribution while all remaining variables were analysed using a lognormal distribution.

For CRS versus TUS, a weighted kappa test was used to assess the agreement between these two screening tests (SAS PROC FREQ, with agree function and weighted kappa test). Model estimates, confidence limits, and standard errors (where applicable) from log- and square root-transformed data were back-transformed for the use in figures and tables. For log-transformed data, the exponential was taken to back-transform estimates, while for standard errors, the exponential of the estimate was multiplied by the standard error to obtain the back-transformed standard error. For square root transformed data, the square of the estimate was used to back transform the estimate, while the estimate*2 was multiplied by the standard error to obtain the back-transformed standard error. Raw data was used to present data comparing CRS versus TUS.

## Ethical approval

The experiment was approved by the Teagasc Animal Ethics Committee (Fermoy, Ireland, Approval number: TAEC2021−326) and the Health Products Regulatory Authority (Dublin, Ireland, Approval number: AE19132/P154).

## Results

### Descriptive statistics

**TUS.** A total of 325 TUS were recorded from a total of 65 calves on five occasions including pre-transport (T1), arrival (T2), 3 or 7 days post arrival (T3), 9 or 10 days post arrival (T4), and 18 days post arrival (T5). No calves died during the transport or during the three weeks thereafter. The distribution of TUS for every time point and the change of TUS per calf between time points (alluvial plot) is shown in Fig 2. Out of 65 enrolled calves, 67% (44/65) had ultrasound findings suggestive of lung pathology (TUS score ≥ 2) on at least one occasion, 28% (18/65) had a severe TUS (score = 3 or 4) on at least one occasion. Out of all severe scores, 61% (20/33) were recorded as consolidation solely in the cranial aspect of the right lung, while one severe score was based on consolidation located in the caudal aspect of the right lung, and one in the caudal aspect of the left lung. One severe score was based on consolidation in the right caudal and cranial lung aspects, while the remaining seven were based on multiple relatively severe lobular lesions in both cranial and caudal right and left lung lobes. Only four calves maintained a TUS of 0 across all time points, while a TUS of 4 was observed in four different calves on single occasions.

The number of calves with a higher, equal, or lower TUS at a subsequent time point compared to the previous time point is presented in Fig 3. Descriptively, the pattern of change transitioned from more deterioration in TUS between T1 and T4 (more "higher" scores), to a recovery pattern between T4 and T5 (more "lower" scores). Four calves had deteriorated TUS for two consecutive time points while three calves gradually recovered based on TUS for two consecutive time points, no calves showed gradual worsening or improvement of TUS for more than two consecutive time points. Four calves maintained a TUS of 0 throughout.

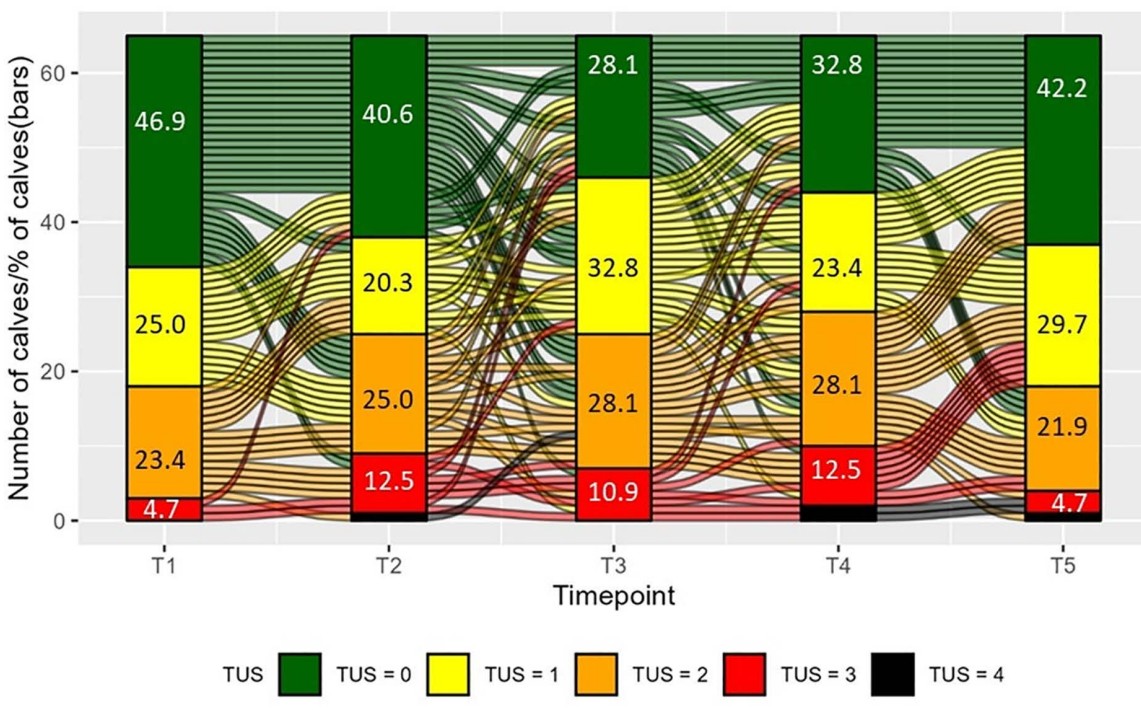

**Fig 2. Change (alluvial lines) of thoracic ultrasound scores (TUS, from 0: healthy, to 4: severe respiratory disease signs) per calf between time points (T1: pre-transport, T2: Arrival, T3: 1-week post arrival, T4: 2-weeks post arrival, and T5: 3-weeks post arrival) and distribution of TUS (vertical bars, % of calves) per time point.** Lines between time point-blocks show changes in the severity of TUS of individual calves.

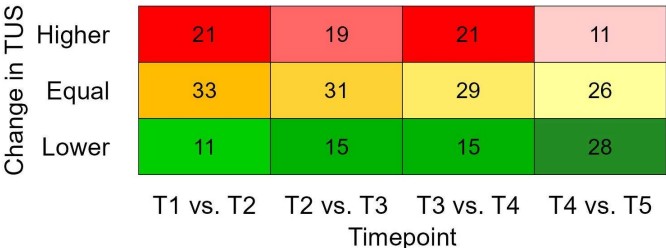

**Fig 3. Number of calves with a higher, equal, or lower thoracic ultrasound score (TUS) at a subsequent time point compared to the previous time point (T1: pre-transport, T2: Arrival, T3: 1-week post arrival, T4: 2-weeks post arrival, and T5: 3-weeks post arrival).** Frequencies are indicated with colours ranging from dark to light within a row for higher values (reds; "more pathology"), equal values (yellows, "equal pathology"), and lower values (greens; "less pathology").

**CRS.** A total of 322 CRS were recorded from a total of 65 calves across five time points (T1 – T5). The distribution of CRS for every time point (vertical bars) and the change of CRS per calf between time points (alluvial lines) is shown in Fig 4. Clinical respiratory scores were lowest at T1, with a maximum CRS of 1 recorded for three different calves, severe CRS (CRS = 3) appeared for two calves per time point at T3, T4, and T5, none of these calves had a severe CRS on more than one time point. On average, 41% of the total sum of CRS component scores came from elevated temperatures, although only 13 observations (out of 322) recorded a score of 3 for elevated temperature. Of the remaining CRS totals, 32% originated from eye discharge,

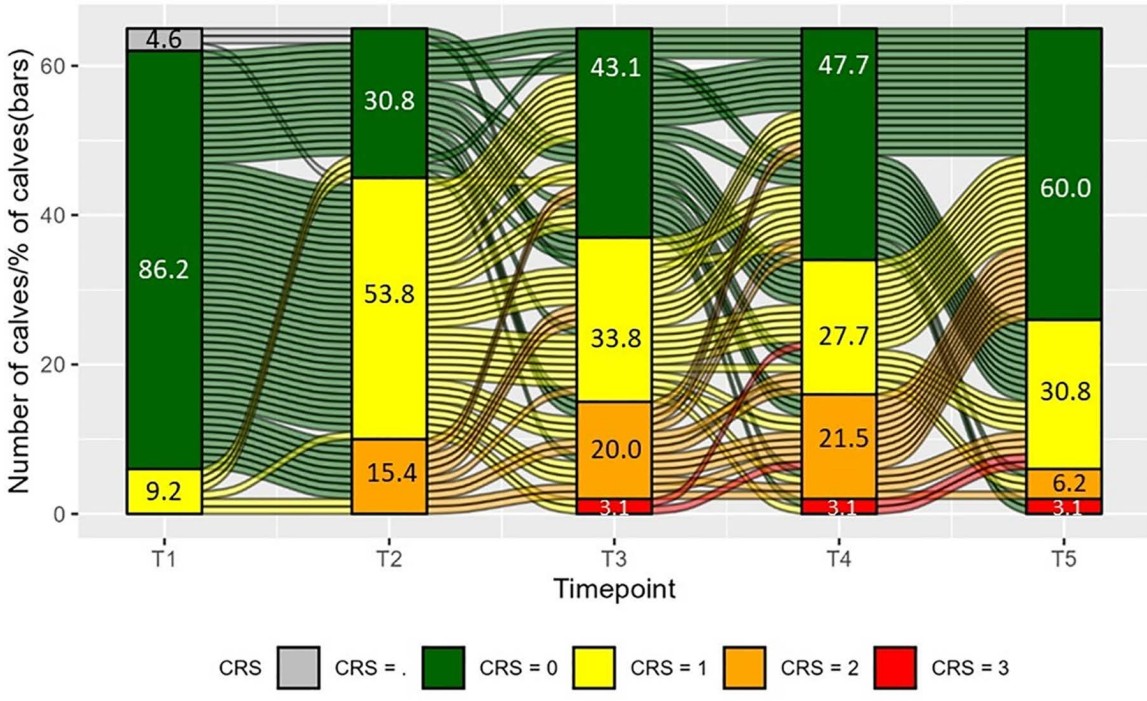

**Fig 4. Change (alluvial lines) of clinical respiratory scores (CRS: from 0: healthy, to 3: severe clinical respiratory disease signs) per calf between time points (T1: pre-transport, T2: Arrival, T3: 1-week post arrival, T4: 2-weeks post arrival, and T5: 3-weeks post arrival) and distribution of TUS (vertical bars, % of calves) per time point.** CRS = . indicates missing values (grey bar). Lines between time point-blocks show changes in the severity of CRS of individual calves.

12% originated from nose discharge, and 10% and 5% originated from altered ear position and coughing respectively. No calves maintained a CRS of 0 across all time points.

The number of calves with a higher, equal, or lower CRS at a subsequent time point compared to the previous time point is shown in Fig 5. Descriptively, the pattern of change transitioned from more deterioration in CRS between T1 and T2 (more "higher" scores), to more signs of recovery between T3 and T4 (more "lower" scores), while deteriorations and recoveries were similar between T4 and T5. Eighteen calves deteriorated in CRS for two consecutive time points and two calves deteriorated in CRS for three consecutive time points, fifteen calves showed recovery based on CRS for two consecutive time points but no calves consecutively showed recovery based on CRS for more than two time points.

### Blood variables

In total, 260 blood samples were collected from 65 calves across four time points, some blood samples were unable to be analysed for some variables, particularly for total immunoglobulins for which 18 out of 260 observations yielded samples too small for analysis. Values (Estimate ± SEM) of blood variables per time point are presented in Table 4. Twelve out of fourteen variables changed between time points. This includes a decrease in haemoglobin and neutrophil, lymphocyte, and eosinophil counts between T1 and T2, while neutrophil/lymphocyte ratio decreased after T4. White blood cell count, monocyte count, and serum amyloid-A gradually decreased between T1 and T5 but showed no rapid changes between consecutive time points. Red blood cell count was highest at T4. Basophil count and Immunoglobulin-A increased between T2 and T4, but were not different at T1 or T5.

### Comparison of TUS, CRS, and blood variables

**TUS vs. blood variables.** Values of blood variables by TUS and effect of TUS on blood variables are presented in Table 5. Lymphocyte count, serum amyloid-A, total immunoglobulins, immunoglobulin-G, and immunoglobulin-M changed based on TUS severity ($p \leq 0.05$). Lymphocyte count was lower for a TUS ≥ 3 than for a TUS of 1 or 2 (3.48 vs. 4.43 and 4.36 $10^9$/L; both $p < 0.05$). Serum amyloid-A was lower for a TUS of 0 than for a TUS of 2 or ≥ 3 (42.1 vs. 65.4 and 73.3 mg/L; both $p < 0.05$). Total immunoglobulins were lower for a TUS ≥ 3 than for a TUS of 2 (10.7 vs. 16.0 g/L; $p = 0.03$). Similarly, immunoglobulin-G was lower for a TUS ≥ 3 than for a TUS of 2 (10.0 vs. 15.2; $p = 0.04$). Immunoglobulin-M was lower for a TUS of 0 than for a TUS of 1 or 2 (423 vs. 508 and 524 mg/L; both $p < 0.05$).

**CRS vs. blood variables.** Values of blood variables by CRS and effects of CRS on blood variables are presented in Table 6. Haemoglobin, white blood cel count, lymphocyte count, and eosinophil count changed based on CRS severity ($p \leq 0.05$). Haemoglobin was lower for a CRS of 1 or 2 than for a CRS of 0 (4.57 and 4.36 vs 5.30 mmol/L; both $p < 0.01$). Similarly, white blood cell count was lower for a CRS of 1 or 2 than for a CRS of 0 (8.71 and 8.24 vs 9.52 mmol/L; both

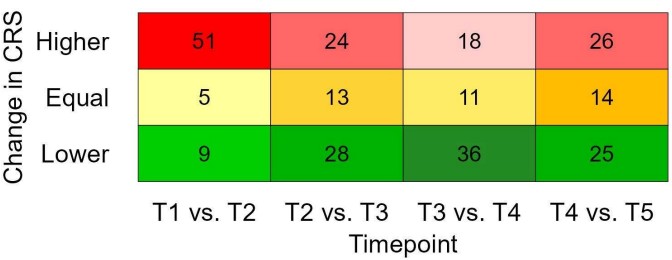

**Fig 5. Number of calves with a higher, equal, or lower clinical respiratory score (CRS) at a subsequent time point compared to the previous time point (T1: pre-transport, T2: Arrival, T3: 1-week post arrival, T4: 2-weeks post arrival, and T5: 3-weeks post arrival).** Frequencies are indicated with colours ranging from dark to light within a row for higher values (reds; "more pathology"), equal values (yellows, "equal pathology"), and lower values (greens; "less pathology").

**Table 4. Values (Estimate±SEM) of blood variables indicative of respiratory disease in calves per time point (T1: pre-transport, T2: Arrival, T3: 1-week post arrival, T4: 2-weeks post arrival, and T5: 3-weeks post arrival) and the effect of Time on individual blood variables.**

| Variable | Timepoint | | | | |
|---|---|---|---|---|---|
| | T1 | T2 | T4 | T5 | p-value |
| RBC ($10^{12}$/L) | 9.70±0.184[a] | 9.81±0.186[a] | 10.13±0.214[b] | 9.73±0.208[a] | < 0.01 |
| HGB (mmol/L) | 6.84±0.178[a] | 4.80±0.180[b] | 4.81±0.176[b] | 4.54±0.161[c] | < 0.01 |
| WBC ($10^9$/L) | 10.68±0.369[a] | 8.38±0.298[bc] | 8.40±0.256[b] | 9.24±0.220[c] | < 0.01 |
| Neutrophil ($10^9$/L) | 4.06±0.269[a] | 3.05±0.202[b] | 2.83±0.168[b] | 2.62±0.111[b] | < 0.01 |
| Lymphocyte ($10^9$/L) | 21.31±0.225[a] | 13.60±0.132[b] | 16.73±0.134[c] | 25.47±0.192[a] | < 0.01 |
| N/L ratio | 0.89±0.083[a] | 0.86±0.070[a] | 0.72±0.045[a] | 0.54±0.030[b] | < 0.01 |
| Monocyte ($10^9$/L) | 0.90±0.073[a] | 1.17±0.051[bc] | 1.05±0.039[ab] | 1.19±0.042[c] | < 0.01 |
| Basophil ($10^9$/L) | 0.23±0.035[ab] | 0.20±0.009[a] | 0.26±0.012[b] | 0.24±0.022[ab] | < 0.01 |
| Eosinophil ($10^9$/L) | 0.08±0.008[a] | 0.04±0.005[b] | 0.02±0.002[c] | 0.02±0.002[c] | < 0.01 |
| SAA (mg/L) | 61.4±6.2[a] | 80.4±6.3[a] | 30.6±3.4[b] | 20.7±2.7[c] | < 0.01 |
| Total IG (g/L) | 13.9±1.17 | 14.4±1.22 | 14.7±1.23 | 12.8±1.08 | 0.54 |
| IgA (mg/L) | 92.2±6.41[ab] | 98.0±8.53[a] | 81.9±5.81[b] | 85.1±5.65[ab] | 0.02 |
| IgG (g/L) | 13.0±1.16 | 13.7±1.22 | 13.9±1.23 | 12.1±1.07 | 0.56 |
| IgM (mg/L) | 398±25.2[a] | 449±30.5[ab] | 546±29.1[c] | 525±26.3[bc] | < 0.01 |

Mean values without a common superscript (a, b, c, d) differ significantly (p<0.05).

**Table 5. Values of blood variables indicative of respiratory disease in calves for thoracic ultrasound scores (TUS) 0, 1, 2, and ≥ 3 (0 is considered healthy and ≥ 3 is considered to have severe respiratory disease signs) and the effect of TUS on blood variables.**

| | TUS | | | | |
|---|---|---|---|---|---|
| | 0 | 1 | 2 | 3 | p-values |
| RBC ($10^{12}$/L) | 9.82±0.200 | 9.81±0.203 | 9.90±0.206 | 9.87±0.238 | 0.90 |
| HGB (mmol/L) | 5.23±0.207 | 4.96±0.214 | 4.78±0.213 | 4.53±0.293 | 0.14 |
| WBC ($10^9$/L) | 9.10±0.287 | 9.29±0.330 | 9.07±0.333 | 9.01±0.494 | 0.93 |
| Neutrophil ($10^9$/L) | 3.01±0.168 | 3.22±0.212 | 2.94±0.198 | 3.52±0.364 | 0.39 |
| Lymphocyte ($10^9$/L) | 4.23±0.174[ab] | 4.43±0.197[a] | 4.36±0.202[a] | 3.48±0.263[b] | 0.01 |
| N/L ratio | 0.72±0.050 | 0.74±0.060 | 0.68±0.056 | 0.98±0.124 | 0.08 |
| Monocyte ($10^9$/L) | 1.01±0.046 | 1.08±0.060 | 1.14±0.065 | 1.18±0.105 | 0.23 |
| Basophil ($10^9$/L) | 0.24±0.019 | 0.20±0.019 | 0.26±0.026 | 0.22±0.033 | 0.19 |
| Eosinophil ($10^9$/L) | 0.04±0.004 | 0.03±0.005 | 0.03±0.004 | 0.03±0.006 | 0.34 |
| SAA (mg/L) | 42.1±4.21[a] | 49.9±5.93[ab] | 65.4±6.84[b] | 73.3±11.28[b] | < 0.01 |
| Total IG (g/L) | 13.4±1.04[ab] | 14.2±1.25[ab] | 16.0±1.44[a] | 10.7±1.45[b] | 0.04 |
| IgA (mg/L) | 83.5±6.05 | 87.6±6.73 | 95.5±7.51 | 104.2±11.11 | 0.15 |
| IgG (g/L) | 12.7±1.03[ab] | 13.5±1.25[ab] | 15.2±1.44[a] | 10.0±1.42[b] | 0.05 |
| IgM (mg/L) | 423±25.4[a] | 508±32.8[b] | 524±34.8[b] | 521±49.2[ab] | < 0.01 |

Mean values without a common superscript ([a], [b]) differ significantly (p<0.05).

p<0.05) and lymphocyte count was lower for a CRS of 1 or 2 than for a CRS of 0 (4.07 and 3.85 vs 4.57 $10^9$/L; both p<0.01). Eosinophil count did not show differences in the least-square means comparisons of CRS, despite an overall effect of CRS (p=0.05). Red blood cell count was not within the absolute significance level (p=0.0504), and least-square means comparisons were never different (all p>0.1).

**Table 6. Values of blood variables for clinical respiratory scores (CRS) 0, 1, 2, and 3 (0 is considered healthy and 3 is considered to be severe clinical signs of disease) and the effect of CRS on blood variables.**

| | CRS | | | | |
|---|---|---|---|---|---|
| | **0** | **1** | **2** | **3** | *p*-values |
| RBC (10$^{12}$/L) | 9.79±0.195 | 9.88±0.200 | 9.95±0.224 | 10.60±0.370 | 0.05 |
| HGB (mmol/L) | 5.30±0.187[a] | 4.57±0.181[b] | 4.36±0.244[b] | 4.75±0.610[ab] | < 0.01 |
| WBC (10$^9$/L) | 9.52±0.262[a] | 8.71±0.280[b] | 8.24±0.400[b] | 8.95±1.050[ab] | < 0.01 |
| Neutrophil (10$^9$/L) | 3.15±0.153 | 3.00±0.180 | 3.06±0.294 | 2.55±0.618 | 0.75 |
| Lymphocyte (10$^9$/L) | 4.57±0.157[a] | 4.07±0.177[b] | 3.85±0.255[b] | 4.62±0.594[ab] | < 0.01 |
| N/L ratio | 0.72±0.044 | 0.77±0.057 | 0.82±0.096 | 0.57±0.166 | 0.47 |
| Monocyte (10$^9$/L) | 1.05±0.040 | 1.15±0.055 | 1.08±0.085 | 1.36±0.275 | 0.27 |
| Basophil (10$^9$/L) | 0.23±0.015 | 0.24±0.021 | 0.22±0.032 | 0.27±0.103 | 0.95 |
| Eosinophil (10$^9$/L) | 0.04±0.003 | 0.03±0.003 | 0.03±0.006 | 0.01±0.010 | 0.07 |
| SAA (mg/L) | 39.2±3.33 | 44.4±5.11 | 46.8±9.00 | 88.3±44.78 | 0.34 |
| Total IG (g/L) | 14.0±0.95 | 14.0±1.13 | 13.4±1.68 | 16.0±4.92 | 0.96 |
| IgA (mg/L) | 86.6±5.86 | 92.6±6.81 | 95.7±9.39 | 96.2±20.46 | 0.49 |
| IgG (g/L) | 13.2±0.94 | 13.1±1.12 | 12.7±1.67 | 15.2±4.95 | 0.96 |
| IgM (mg/L) | 469±25.2 | 484±29.3 | 521±44.6 | 484±94.1 | 0.61 |

Mean values without a common superscript ([a, b]) differ significantly (p<0.05).

**TUS vs CRS.** A heat map of the distribution of TUS (0–3) and CRS (0–3) is presented in Fig 6. Comparatively, of the calves with a CRS of 0 at T1 (86.2% of total), 30% had TUS of 2 or 3. Of the calves with a TUS of 0 at the same time point (46.9% of total), only 10% had a CRS of 1 and higher CRS were not observed. A weighted Cohen's kappa test yielded a value of 0.038 (95% CI: −0,0338–0,1100), indicating an agreement close to zero. Severe TUS (TUS ≥ 3) never correlated to a severe CRS (CRS = 3).

## Discussion

Early detection of respiratory disease in calves is crucial to guarantee calves are fit for travel and to improve treatment outcomes and calf welfare. In this study, we compared three screening tests (TUS, CRS, and blood variables) for the assessment of potential respiratory dysfunction in calves before and after transport, to assess the degree of agreement in identifying respiratory dysfunction in transported calves.

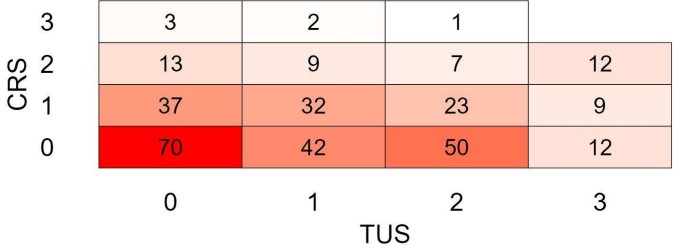

**Fig 6. Heat map of frequency of thoracic ultrasound scores (TUS, from 0: healthy, to 3: severe respiratory disease signs) vs. clinical respiratory scores (CRS, from 0: healthy to 3: severe clinical respiratory disease signs) for all observations at all time points in calves transported from Ireland to The Netherlands.**

## Overview of respiratory disease prevalence for all screening methods

Long distance transport of animals is frequently linked to increased incidence of respiratory disease. The reasons for this have been well described [23] and take root from the increase in pathogen exposure from mixing of animals, increase in stress and decrease in regular and sufficient feeding. An increase in respiratory disease can therefore be anticipated during and following transport; such a change was detected by all three diagnostic techniques, but the extent of this change was not comparable.

### TUS

The thoracic ultrasonic approach, the only one of the three techniques which examined the lungs directly, revealed that only 47% of calves selected for transport in Ireland (and deemed fit for transport by exporters and veterinarians) had unaffected lung tissue. The fact that pre-transport ultrasound scanning revealed abnormalities in the lungs of almost 50% of calves (with one in four with moderate or severe lesions) questions the sensitivity of current pre-transport health assessments, such as clinical respiratory scoring [6]. Pre-transport, 4.7% of calves had a TUS $\geq 3$, which is identical to on-farm prevalence for Irish dairy calves [24]. Overall, the deviation from health in transported calves peaked at approximately 7 days after transport, as indicated by the fact that 72% of study calves displayed a TUS $\geq 1$ at T3, despite antibiotic interventions during this period. A partial recovery was observed by 21 days with 42% of calves with healthy lungs according to TUS analysis.

While four calves maintained a healthy score throughout, it was more common for the TUS, on an individual level, to continually change. For example, from a total of 26 calves with a severe TUS up to T4, six no longer showed any symptoms of respiratory disease (TUS $= 0$) at a later time point. This suggests a potential for ultrasound lesions to resolve over time with antibiotic intervention. While there were good signs of recovery in some calves, not all calves recovered fully, despite receiving three courses of antibiotics in the first three weeks on the veal farm. The degree of change in TUS was variable, but continued deterioration was rare, with only two calves demonstrating an increase in TUS for two consecutive time points. Transient lesions may partially explain changes in TUS between timepoints, but further analysis is required to explore the exact causes of this variability. Understanding changes in the lung pathology is needed to correctly schedule screening for respiratory health to improve the timing of individual treatments.

### CRS and blood

The profile of CRS change over time also followed the anticipated pattern, but the percentage of calves classed as healthy increased or decreased more rapidly, particularly between T1 and T2. It must be noted that CRS calculations pre-transport did not include rectal temperatures. These were omitted from observations due to practical concerns; however, even when omitting rectal temperatures from CRS calculations at all time points, the general pattern of change remained similar, with a rapid increase in CRS values between T1 and T2 and a gradual decrease thereafter. Regardless, the incomplete calculation of CRS pre-transport may have underestimated disease. The time gap between T1 and T2 (three to four days after the initial transport-related pathogen exposure) is within the limits for observing the clinical signs of infection, while accounting for the exacerbation in CRS values due to the rigors of transport.

Changes in blood variables over time are hard to evaluate in the absence of sensitive cut-off values, though most changes appeared between T1 and T2, and can be assumed to be the result of immune responses to early pathogen exposure compounded by transport stress [13,25].

### Comparison of TUS, CRS, and blood variables

The degree of agreement between the three diagnostic techniques on the severity of respiratory disease was poor, particularly so between TUS and CRS. Clinical respiratory scoring is the most employed method for on-farm health screenings

in calves by farmers and veterinarians, while veterinarians may employ further screening including thoracic auscultation or ultrasound if clinical signs of respiratory disease are present [6,26].

**CRS vs. TUS.** Both CRS and TUS are designed to identify calves with respiratory dysfunction, albeit with different modalities. In this study, there was almost no agreement between these methods, highlighting a clear discordance in field assessments. Descriptively, TUS detected respiratory abnormalities in 60% of calves deemed clinically healthy (based on a normal CRS), while 29% had moderate lesions and 7% had severe lesions. This was also confirmed when comparing changes in TUS and CRS over time; improvement in CRS generally occurred between T3 and T4, while recoveries of TUS peaked between T4 and T5. To some degree, TUS may capture early infections, prior to the onset of clinical signs. However, when severe consolidation is present, clinical signs would be expected. Some clinical cases without synchronous ultrasound findings may be caused by upper respiratory infections, which are not visible during thoracic ultrasound scanning. In general, these results indicate that CRS underestimates early-onset lung lesions that TUS can detect. The discordance also has practical implications: relying solely on CRS could lead to under-identifying of at-risk calves for treatment or fitness for transport assessments.

**TUS and CRS vs blood.** Discordant results of TUS and CRS have previously been reported [24], but the comparison with blood variables in this study was novel. Comparisons between TUS and blood variables were more indicative and clear relationships between TUS severity and blood indicators of immunity and inflammation were evident. Comparisons between CRS and blood variables were mildly indicative of changes in the immune system and oxygen-carrying capacity of the calf, but on a variable-by-variable overview, fewer variables correlated with CRS than with TUS.

Haemoglobin can increase or decrease in response to severe respiratory disease [13,27]. Haemoglobin decreased as a result of clinical symptoms in our study, though this difference was merely numerical for comparisons between TUS and blood variables. Blood immune variables were more consistently correlated with other screening methods, particularly when compared to TUS. Lymphopenia was correlated in both TUS and CRS comparisons. Lymphopenia is commonly found in response to respiratory infections in calves [28–30], and a similar response was observed in our study. Immunoglobulins were particularly sensitive to severe TUS in our study. As shown by Pardon et al. [16], low immunoglobulins at arrival predict development of respiratory disease during the production stage, and high immunoglobulin counts are often seen during respiratory infections [13]. Low total immunoglobulins and IgG in our calves may have been a result of these calves being more at risk of developing lung consolidation, while the increase in IgM in correlation with TUS was possibly indicative of an antibody response to respiratory pathogens. Aside from the immune system, the acute phase protein response was also affected, evidenced by an increase in SAA for moderate to severe TUS. This response was expected and the concentration of SAA in "sick" calves (as evidenced by a TUS of 2 or 3) was similar to that found in calves diagnosed with respiratory disease previously [31,32].

## Practical considerations and implementations

Regardless of the method used, large numbers of calves were identified with respiratory dysfunction, but were able to recover from this disease in most cases, potentially assisted by the use of antibiotics. However, the damage to lung tissue caused by respiratory dysfunction may have negative effects, such as decreased growth and greater susceptibility to further disease, for the rest of the calf's life [33].

From a functional perspective, CRS is an easily learned, quick and costless procedure requiring less than one minute per calf for a full assessment, making it a practically efficient method. However, based on this study, and previous studies unrelated to transport, it is not an effective screening tool with questionable sensitivity and therefore should not be relied upon as unaccompanied screening technique. Its poor relationship to TUS (which typically has better sensitivities and specificities [10,11]) and blood variables underline this. In contrast, TUS yielded results more comparable to immune blood variables and, on the basis of this study could be recommended as a screening test due to its instant and sensitive results that show the severity and locations of the lesions and is less impacted by external factors such as stress.

Functionally, TUS is a time-consuming method, particularly when a complete analysis of all aspects of both lungs is undertaken, requiring approximately four minutes per calf for a trained observer [34]. This point, and the added cost and training requirement relative to CRS must be balanced against the overall benefit of this screening technique. Some of these practical limitations of TUS could be mitigated to a degree. For example, a single scan of the right middle to cranial lobes and the left cranial lobe may be sufficient when screening a cohort of calves for respiratory infections [35]. In the case of our study, scanning the right cranial lobe exclusively would have captured 61% of severe scores (TUS ≥ 3). This strategy must be validated but could potentially reduce the time to result by up to 75%, and in turn enhance the selection of calves fit for transport. It is also plausible that routinely incorporating TUS as a screening tool may subconsciously re-train observers in their CRS assessments.

Blood variables have the benefit of objectivity, though blood sample collection and analyses are time consuming. More work is needed to determine cut-off values for blood variables and to investigate whether fast on-site analysis may improve the value of blood variables in respiratory disease detection. External factors such as stress, a common occurrence during transport, impact blood values and can vary above and below normal during the course of a single infection, which limits the accuracy of blood as an indicator of respiratory disease.

### Limitations

Firstly, the batch antibiotic treatments in this study are likely to have limited disease progressions, particularly limiting clinical signs of disease. Batch antibiotic treatments are standard practice in veal systems, and therefore the presentation of the current data is still representative for calves transported long-distance to intensive veal systems. Early treatment is also likely why no calves died, and may have reduced the occurrence of severe respiratory disease in the study calves. Non-synchronous sampling of TUS/CRS and bloods likely reduced correlations between bloods and observation-based screening tests. However, severe infections have a gradual infection pattern and changes in blood values in response to severe infection would be expected to be measurable for several days. Pre-transport assessments were performed one-day earlier for calves at the dairy farm than for mart calves, but the fact these assessments were performed pre-transport; before exposure to commingling and transport, meant that they were deemed sufficiently equal to be assessed as a single timepoint. Thoracic ultrasonography and CRS data in this trial were collected in suboptimal situations; the farm environment, in which calves were housed individually, made TUS observations complicated due to limited space for the observer in the pen next to the calf. The pooling of TUS scores 3 and 4 was necessary due to a low number of observations of TUS 4. TUS classification in calves is often based on scores of 3 or above, or based on lesions of more than 1 or 3 cm in diameter [6]. Therefore, the pooling of these scores in the current paper still represents a clear line in physiological differences. Finally, this study itself increased the stress on the calves due to increased handling, which may have affected the results.

### Conclusion

This study found a high prevalence of lung lesions in calves at all time points. Respiratory disease screening based on TUS, CRS, and blood variables did not align consistently, though comparisons between TUS and blood variables were more indicative than comparisons between CRS and blood variables. Lymphocyte count was lower for calves with severe TUS and moderate CRS, but no other blood variables consistently changed in conjunction with TUS and CRS. Despite limited comparative options, the ability of TUS to identify the severity of thoracic lesions in individual calves should be taken into account. TUS showed better alignment with immune markers than CRS, supporting its use in high-risk settings. Further research should explore causes of high respiratory disease prevalence and should assess whether pre-transport TUS could be used to assess the odds of respiratory disease development post-transport.

## Supporting information

**S1 File. The dataset supporting the findings of this study, including all variables and metadata necessary to replicate the results, is provided with this submission as supplementary File S1.**
(XLSX)

## Acknowledgments

We would like to thank the commercial exporters and representatives of the VanDrie Group for their support and for providing facilities, and Rimondia B.C. (Elspeet, the Netherlands) for their help with blood sampling and analysis in the Netherlands. Our sincere thanks go to Emma Gregorio, Julie Lane, Pat Dillon, Laurence Shalloo, Chloe Millar, Tomas Tubritt, Hazel Costigan, Fionnuala McDermott, Des Lane, Charles Dwan, and Natasha Browne (Teagasc Moorepark, Fermoy, Ireland) for joining us during transport and for all their help with the many samples collected in Ireland and France. We're also very grateful to Francesca Marcato, Maaike Wolthuis-Fillerup, and Henk Gunnink (Wageningen University and Research, Wageningen, the Netherlands) for their help with sampling and processing in the Netherlands, and to Margaret Murray (Teagasc Grange, Dunsany, Ireland) for processing and analysing all blood samples.

## Author contributions

**Conceptualization:** Luca L. van Dijk, Susanne Siegmann, Niamh L. Field, Cornelis G. van Reenen, Eddie A.M. Bokkers, Gearoid Sayers, Muireann Conneely.

**Data curation:** Luca L. van Dijk, Susanne Siegmann, Niamh L. Field, Katie Sugrue, Cornelis G. van Reenen, Eddie A.M. Bokkers, Gearoid Sayers, Muireann Conneely.

**Formal analysis:** Luca L. van Dijk, Niamh L. Field, Gearoid Sayers.

**Funding acquisition:** Niamh L. Field, Cornelis G. van Reenen, Eddie A.M. Bokkers, Gearoid Sayers, Muireann Conneely.

**Investigation:** Niamh L. Field, Cornelis G. van Reenen, Eddie A.M. Bokkers, Gearoid Sayers, Muireann Conneely.

**Methodology:** Luca L. van Dijk, Niamh L. Field, Cornelis G. van Reenen, Gearoid Sayers, Muireann Conneely.

**Project administration:** Niamh L. Field, Katie Sugrue, Muireann Conneely.

**Supervision:** Niamh L. Field, Gearoid Sayers, Muireann Conneely.

**Visualization:** Luca L. van Dijk.

**Writing – original draft:** Luca L. van Dijk.

**Writing – review & editing:** Susanne Siegmann, Niamh L. Field, Katie Sugrue, Cornelis G. van Reenen, Eddie A.M. Bokkers, Gearoid Sayers, Muireann Conneely.

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
