## [Decision Letter · Decision Letter 0]

29 Aug 2025

PONE-D-25-22875Comparison of thoracic ultrasonography (TUS), clinical respiratory scoring (CRS), and blood analysis to evaluate respiratory dysfunction in transported calvesPLOS ONE

Dear Dr. van Dijk,

Thank you for submitting your manuscript to PLOS ONE. After careful consideration, we feel that it has merit but does not fully meet PLOS ONE’s publication criteria as it currently stands. Therefore, we invite you to submit a revised version of the manuscript that addresses the points raised during the review process.

Following a careful assessment by the editorial team and a group of expert reviewers, we have determined that the paper shows significant potential for publication. However, based on the reviewers' feedback, the manuscript requires MAJOR REVISIONS to meet the journal's standards. Please see the enclosed reviewer comments for detailed suggestions and specific actions required to improve the manuscript. We invite you to revise and resubmit the manuscript to us for further consideration. We look forward to receiving the revised manuscript.

We look forward to receiving your revised manuscript.

Kind regards,

Mourad Mahmoud

Academic Editor

PLOS ONE

**Journal Requirements:**

1. When submitting your revision, we need you to address these additional requirements. Please ensure that your manuscript meets PLOS ONE's style requirements, including those for file naming. The PLOS ONE style templates can be found at https://journals.plos.org/plosone/s/file?id=wjVg/PLOSOne_formatting_sample_main_body.pdf and https://journals.plos.org/plosone/s/file?id=ba62/PLOSOne_formatting_sample_title_authors_affiliations.pdf 2. To comply with PLOS ONE submissions requirements, in your Methods section, please provide additional information regarding the experiments involving animals and ensure you have included details on (a) methods of sacrifice, (b) methods of anesthesia and/or analgesia, and (c) efforts to alleviate suffering. 3. We noted in your submission details that a portion of your manuscript may have been presented or published elsewhere. “The blood data presented in this paper has previously been reported in two separate papers (attached as Paper 1 and Paper 2), and the CRS data has been reported in Paper 2 in a different format. However, the current paper is the first to incorporate TUS data, and it is the only one in which multiple methods are directly compared. This paper also uses different data analysis approaches and reporting structures, and therefore constitutes a distinct and independent publication.”  Please clarify whether this [conference proceeding or publication] was peer-reviewed and formally published. If this work was previously peer-reviewed and published, in the cover letter please provide the reason that this work does not constitute dual publication and should be included in the current manuscript. 4. Please include captions for your Supporting Information files at the end of your manuscript, and update any in-text citations to match accordingly. Please see our Supporting Information guidelines for more information: http://journals.plos.org/plosone/s/supporting-information. 5. If the reviewer comments include a recommendation to cite specific previously published works, please review and evaluate these publications to determine whether they are relevant and should be cited. There is no requirement to cite these works unless the editor has indicated otherwise.

Reviewers' comments:

Reviewer's Responses to Questions

**Comments to the Author**

1. Is the manuscript technically sound, and do the data support the conclusions?

Reviewer #1: Partly

Reviewer #2: Yes

2. Has the statistical analysis been performed appropriately and rigorously? 

Reviewer #1: No

Reviewer #2: Yes

3. Have the authors made all data underlying the findings in their manuscript fully available?

Reviewer #1: No

Reviewer #2: Yes

4. Is the manuscript presented in an intelligible fashion and written in standard English?

Reviewer #1: Yes

Reviewer #2: Yes

5. Review Comments to the Author

**Reviewer #1:** TUS and CRS comparisons with blood are made despite blood being sampled on different days than clinical assessments (T2, T4, T5). This can lead to inconsistencies and should be addressed with caution or sensitivity analysis.

Clarify this limitation in the methods more explicitly and consider whether statistical correction or subgroup analysis might mitigate this mismatch.

Some values were excluded due to suspected lab errors (e.g., IgM, eosinophils). However, no mention is made of imputation or sensitivity analyses.

The kappa value (0.038) indicates no meaningful agreement, yet this is only discussed briefly. This result is central to the paper's conclusions

Grouping TUS score 4 into score 3:

• This collapsing may affect results significantly given the small number of TUS 4 observations. A sensitivity analysis keeping TUS 4 separate, or pooling with discussion of clinical relevance, would improve rigor.

Use more concise language in the abstract and results.

“...followed by a gradually decrease” → should be “a gradual decrease”

“...severe signs of respiratory disease” → could be simplified to “severe respiratory signs”

"The CRS system was based on the Wisconsin calf health scorer..." → should be “based on the Wisconsin Calf Health Scoring Chart”

Ensure past tense is used consistently in the methodology (e.g., "calves were scored" vs. "calves are scored")

Several sections (especially Discussion) contain overly long paragraphs. Breaking these into thematic blocks would improve clarity.

This is a scientifically sound manuscript with good potential. However, I recommend a major revision focusing on:

• Statistical robustness and discussion of multiple comparisons.

• Clearer language and grammar corrections.

• Enhanced explanation of mismatched sample timing and potential biases.

• A clearer discussion of the CRS vs. TUS discordance and implications for field diagnosis.

**Reviewer #2:**  Comparison of thoracic 1 ultrasonography (TUS), clinical 2 respiratory scoring (CRS), and 3 blood analysis to evaluate 4 respiratory dysfunction in 5 transported calves

1 . L31 | "TUS scores (0: healthy to 3: severe consolidation)" – Incomplete scale; Table 2 shows scores up to 5. | Revise to: "TUS scores (0: healthy to 5: severe pneumonia)" and clarify in text that scores 4–5 were rare and collapsed into score 3 for analysis (L196–197). |

2 . L32 | "CRS (0: healthy to 3: severe clinical symptoms)" – Implies linear scale, but scoring is categorical. | Clarify: "CRS categories (0: healthy, 1: mild, 2: moderate, 3: severe)" to reflect ordinal nature.

3 . L35 "3 or 7 days post-arrival (T3)" – Introduces variability without justification. | State: "T3 was defined as 3 or 7 days post-arrival due to logistical constraints; analysis adjusted for time."

4 . L41 "Total immunoglobulins and IgG were lower for TUS of 3" – IgG is part of total Ig; cannot be lower unless others decrease. | Recheck calculation: Total Ig = IgA + IgG + IgM. If IgG ↓ and total Ig ↓, clarify if IgA/IgM compensate.

5 . L59 "Calves that are diseased pre-transport should not travel" – Assumes causality not tested. Temper: "Calves with clinical signs pre-transport may be at higher risk during transport."

6 . L65 "CRS has low sensitivity (0.27–0.64)" – No citation for these values in transport context. | Cite source specifically for transport or pre-transport settings (e.g., Donlon et al. 2023, Ref 21).

7 . L74 "TUS has sensitivities >0.85 and up to 0.94" – Cites [10,11], but [10] is subclinical, [11] lacks necropsy validation. Specify: "Sensitivities vary by population; in subclinical calves, Ollivett et al. [10] reported 0.85–0.94 vs. BAL, not necropsy."

8 . L89–94 Calves from farms and marts mixed without accounting for origin differences. Include "origin (farm vs. mart)" as a covariate in models or stratify analysis.

9 . L95–98 Describes cohort differences (sex, age, breed) but no statistical adjustment. Adjust models for cohort (C1 vs C2) or include as random effect.

10 . L105 "Calves from livestock mart assessed on day of transport" – Timing differs from farm calves (L104: one day pre). Standardize or acknowledge bias in pre-transport assessment timing.

11 . L122–124 All calves received antibiotics upon arrival – confounds disease progression. Acknowledge that metaphylaxis limits natural disease progression and affects TUS/CRS/blood interpretation.

12 . L130–133 TUS and CRS not performed on same day at T4/T5 (blood sampled 1–2 days apart). State: "Non-simultaneous sampling may reduce correlation; results interpreted with caution."

13 . L134–136 No blood at T3, but TUS/CRS done – limits temporal comparison. | Justify gap and avoid claiming integrated assessment at T3.

14 . L147 "Depth of 11cm" – Not validated for calf thorax; may miss deep lesions. Cite justification or note potential for missed consolidations.

15 . L153 "6-tier TUS" – Inconsistent with Table 2 (6-point scale) and text (0–3 used). Correct: "A 6-point TUS scale was used, but scores 4–5 were combined with 3 due to low frequency."

16 . L161–163 | Rectal temperature omitted at T1 and imputed at T4 – introduces bias. Exclude temperature from T1 CRS or use alternative scoring; do not impute without validation.

17 . L167–168 CRS categories based on sum 0–1=0, 2–3=1, etc. – unequal binning. Justify binning or cite Wisconsin system for this grouping.

18 . L177 "26ml blood" – excessive volume for young calves (~28d). Confirm volume is safe (<10% total blood volume); typical calf blood volume ~80 mL/kg. 26 mL from 55kg calf = ~0.5% – acceptable, but state safety.

19 . L182–183 Two hematology analyzers used (Advia 2120, XT-1800i) – may introduce batch effects. Perform cross-calibration or include analyzer as covariate in models.

20 . L187 Hb units converted (g/dL to mmol/L) – conversion factor not cited. Cite: "Hb converted using factor 0.6205 (mmol/L per g/dL) [Ref]."

21 . L195–196 One calf removed due to missing TUS – violates intention-to-treat. Report as missing data and use imputation or sensitivity analysis.

22 . L196–197 TUS 4 reclassified as 3 – alters disease severity. Report both ways: primary analysis with TUS 0–3, secondary with TUS 4 separate.

23 . L198–200 Outliers removed without statistical justification (e.g., >3 SD). Define outlier criteria (e.g., >3 IQR) and report number excluded.

24 . L207–212 "Minimal model" not defined; no random effects for repeated measures. Use mixed models with calf ID as random effect to account for repeated measures.

25 . L212 Residuals tested for normality – but no mention of transformation validation. Show Q-Q plots or Shapiro-Wilk results in supplement.

26 . L214–220 Different transformations per variable – but no consistency in back-transformation. Ensure all SEs and CIs are correctly back-transformed (L224–229 are correct; verify in tables).

27 . L222 Weighted kappa used – appropriate, but no confidence interval reported. Add 95% CI for kappa in text or figure.

28 . L241 "No calves died" – but mortality is relevant to BRD severity. Discuss survival bias: severe cases may have been treated early.

29 . L243 "67% had TUS ≥2" – implies disease, but no validation against gold standard. Use: "67% had ultrasound findings suggestive of lung pathology."

30 . L244–248 Location of lesions described, but not analyzed statistically. Report if right cranial lobe predominance is significant (e.g., chi-square).

31 . L257 "Chi-square; p=0.03" – Chi-square for transition frequencies? Inappropriate. Use McNemar’s test or generalized estimating equations (GEE) for paired ordinal data.

32 . L289 "Chi-square; p<0.01" – same error as L257. Replace with GEE or marginal model for longitudinal CRS changes.

33 . L307 "9 out of 14 variables changed" – cherry-picking; no adjustment for multiple testing. Apply FDR correction (e.g., Benjamini-Hochberg) and report q-values.

34 . L308 "Neutrophil/lymphocyte ratio decreased between T4 and T5" – contradicts Table 4 (T3 missing). Correct: "N/L ratio decreased from T1 to T5" or clarify trend.

35 . L315–316 Table 4: HGB different superscripts (a,b) but p= <0.01 – correct, but T5 lower than T2/T4? Verify post-hoc test: T5 is lowest, should be 'c' if significant.

36 . L322 "Haemoglobin did not show differences despite significant p=0.04" – contradiction. Recheck: p=0.04 is significant; either remove "did not show" or correct p-value.

37 . L323 "Lymphocyte count was lower for TUS of 3 than any lower TUS" – but TUS 0 vs 1 not different? Specify: "significantly lower than TUS 0, 1, and 2".

38 . L324 "N/L ratio higher for TUS 3 vs 0 or 2" – but p=0.03 and 0.01 – significant, but not vs 1? Clarify: "not significantly different from TUS 1".

39 . L325 "SAA higher for TUS 2 or 3 than TUS 1" – but Table 5 shows TUS 1=49.9, TUS 2=65.4, TUS 3=73.3. Correct: "higher for TUS ≥2 than TUS 0 or 1".

40 . L326 "Total Ig lower for TUS 3 than TUS 2" – but TUS 1=14.2, TUS 2=16.4 – TUS 1 not different? State: "significantly lower than TUS 2", not "any lower TUS".

41 . L331 "Haemoglobin did not show differences" – but p=0.04 – error. Correct: "Haemoglobin decreased with increasing TUS (p=0.04), though post-hoc differences were not significant."

42 . L341 "Haemoglobin lower for CRS 1 than 0" – but Table 6 shows CRS 2 and 3 also lower. Say: "lowest at CRS 1 and 2", or recheck pairwise tests.

43 . L344 "Eosinophil count did not show differences despite p=0.05" – p=0.05 is significant. Correct: "marginally significant" or adjust alpha.

44 . L355 "Severe TUS never correlated to severe CRS" – but n may be too small. Add: "though low frequency of severe cases limits inference."

45. L377 "almost 50% had some inflammation" – but TUS 1 may not be pathological. Use: "ultrasound abnormalities", not "inflammation", without histology.

46 . L380 "prevalence of severe lesions 4.8%" – based on observations, not calves. Report per calf: "4.8% of calves had at least one TUS=3 or 4 observation."

47 . L382 "72% of study calves displayed lung lesions" – at what time? T3? Specify: "at T3, 72% of calves had TUS ≥1."

48 . L388 "consolidation to fully recover" – assumes reversibility without histology. Say: "ultrasound lesions resolved", not "fully recover".

49 . L394 "failure of ultrasound technique" – contradicts claim of high sensitivity. Replace: "inter-observer variability or transient lesions may explain changes."

50 . L400 "rectal temperatures omitted... due to practical concerns" – weakens CRS validity. Acknowledge CRS is incomplete pre-transport and may underestimate disease.

51 . L416–418 "174 observations of healthy CRS, 104 not healthy by TUS" – but TUS may detect subclinical disease. Reframe: "TUS detected abnormalities in 60% of clinically healthy calves."

52 . L430 "haemoglobin usually increases" – incorrect; Hb often decreases in BRD due to anemia of inflammation. Correct: "In contrast to some reports, haemoglobin decreased, possibly due to acute phase response."

53 . L439 "Low total Ig and IgG... more at risk" – cites [16], but [16] is about arrival Ig predicting disease. Accurate: "As shown by Pardon et al. [16], low Ig at arrival predicts later BRD."

54 . L463 "TUS requires up to five minutes" – but no data in manuscript supports this. Add: "Based on observer logs, average scan time was X minutes."

55 . L465 "scanning right cranial lobe captures 61% of severe scores" – useful, but not validated. Recommend: "may be sufficient for screening, pending validation."

56 . L476–478 Blood and TUS/CRS not simultaneous – major limitation for correlation. State: "non-synchronous sampling likely attenuated correlations between blood and clinical scores."

57 . L481–482 "suboptimal situations" – but no description of how this affected blinding or consistency. Describe environmental constraints and their potential impact on scoring reliability.

58 . L484 "observers wear respiratory masks" – safety note, not research finding. Move to methods or supplementary.

59 . L485 "study increased stress" – confounder not measured or adjusted for. Measure cortisol or behavior, or acknowledge as limitation.

60 . L487–497 Conclusion: "TUS should be compared to other methods" – already done. Reframe: "TUS showed better alignment with immune markers than CRS, supporting its use in high-risk settings."

6. PLOS authors have the option to publish the peer review history of their article (what does this mean? ). If published, this will include your full peer review and any attached files.

**Do you want your identity to be public for this peer review?** For information about this choice, including consent withdrawal, please see our Privacy Policy .

Reviewer #1: **Yes:** Prof Dr Raheela Akhtar

Reviewer #2: **Yes:** Ali Hussein Aldujaily

---

## [Author Response · Author response to Decision Letter 1]

4 Nov 2025

Dear Dr. Mahmoud,

Thank you for taking the time to consider our manuscript. We have carefully reviewed the comments provided by the reviewers and have revised the manuscript accordingly. Detailed, point-by-point responses to all reviewer comments are included.

In addition, we have addressed a few additional queries related to the journal’s submission requirements, as outlined below.

We have amended the file naming and reviewed the journal requirements once more, we hope the manuscript now conforms to the requirements

2) To comply with PLOS ONE submissions requirements, in your Methods section, please provide additional information regarding the experiments involving animals and ensure you have included details on (a) methods of sacrifice, (b) methods of anesthesia and/or analgesia, and (c) efforts to alleviate suffering.

As this was an observational study, we were not in charge of providing euthanasia or analgesia to study animals. They were housed in a commercial environment. We have included the following statement in the methods to clarify this issue: “No animals were euthanized for this study. Analgesia or euthanasia was provided only at the discretion of the farmer and attending veterinarian, and all handling was performed to minimize stress.”

3) We noted in your submission details that a portion of your manuscript may have been presented or published elsewhere. “The blood data presented in this paper has previously been reported in two separate papers (attached as Paper 1 and Paper 2), and the CRS data has been reported in Paper 2 in a different format. However, the current paper is the first to incorporate TUS data, and it is the only one in which multiple methods are directly compared. This paper also uses different data analysis approaches and reporting structures, and therefore constitutes a distinct and independent publication.”

The following statement has been included in the cover letter: “The blood data presented in this manuscript have been previously reported in two separate peer-reviewed papers (attached as Paper 1 and Paper 2). The CRS data have also been reported in Paper 2, but in a different format. The current manuscript is the first to incorporate thoracic ultrasonography (TUS) data and to directly compare multiple methods within the same study. Additionally, it employs different data analysis approaches and reporting structures. Therefore, this manuscript constitutes a distinct and independent publication and does not represent dual publication.”

4) Please include captions for your Supporting Information files at the end of your manuscript, and update any in-text citations to match accordingly. Please see our Supporting Information guidelines for more information: http://journals.plos.org/plosone/s/supporting-information.

We have included the following statement after the acknowledgements section: “Supplementary information

The raw dataset supporting the findings of this study is provided as Supplementary File S1.”

N/A

Dear Reviewer #1,

We sincerely thank you for the time and effort you have dedicated to reviewing our manuscript. We have carefully considered all your comments and revised the manuscript accordingly. Detailed, point-by-point responses to each comment are provided below. In particular, we have expanded the discussion on the grouping of TUS scores and further elaborated on the observed discordance between TUS and CRS to improve clarity and interpretability. We have attached a “clean” and “tracked changes” version of the document for your convenience.

TUS and CRS comparisons with blood are made despite blood being sampled on different days than clinical assessments (T2, T4, T5). This can lead to inconsistencies and should be addressed with caution or sensitivity analysis.

Clarify this limitation in the methods more explicitly and consider whether statistical correction or subgroup analysis might mitigate this mismatch.

We have included additional text on the limitations of non-synchronous sampling in the discussion (L431).

Ensure past tense is used consistently in the methodology (e.g., "calves were scored" vs. "calves are scored")

We have had another read through and hope we have now changed the methods sufficiently to include only past tense statements. The present tense remains only where a figure is described.

Some values were excluded due to suspected lab errors (e.g., IgM, eosinophils). However, no mention is made of imputation or sensitivity analyses.

Thank you for your comment. We have adapted the exclusion criteria for outliers, and have now excluded values greater than *3SD above or below the mean (L208). The methods section and all resulting estimates have been updated.

Several sections (especially Discussion) contain overly long paragraphs. Breaking these into thematic blocks would improve clarity.

We have included some additional sub-sections in the discussion and hope this will clarify the readability.

L39: “...followed by a gradually decrease” → should be “a gradual decrease”

We think this sentence may have been misread, as it follows on the previous sentence which mentions ”CRS rapidly increased between T1 and T2 followed by a gradual decrease”. We have included an extra comma to clarify.

“...severe signs of respiratory disease” → could be simplified to “severe respiratory signs”

Thank you for your comment. This has been changed throughout the document.

"The CRS system was based on the Wisconsin calf health scorer..." → should be “based on the Wisconsin Calf Health Scoring Chart”

Now L161: We have updated this line.

Grouping TUS score 4 into score 3: This collapsing may affect results significantly given the small number of TUS 4 observations. A sensitivity analysis keeping TUS 4 separate, or pooling with discussion of clinical relevance, would improve rigor.

We have tried to analyse the data with TUS 3 and 4 kept separate, but due to the low number of observations of TUS 4, this did not provide additional relevance. We have included additional discussion on the clinical relevance in the limitations section of the discussion.

The kappa value (0.038) indicates no meaningful agreement, yet this is only discussed briefly. This result is central to the paper's conclusions

We have included additional discussion on this topic. (Line 423 - 435)

Include a clearer discussion of the CRS vs. TUS discordance and implications for field diagnosis.

Now L423 – 435: We have expanded the discussion on the TUS vs. CRS comparison.

Dear Reviewer #2

We sincerely thank you for the time and effort you have dedicated to reviewing our manuscript. We have carefully considered all your comments and revised the manuscript accordingly. In particular, the statistical analyses have been updated to include a repeated effect for the blood measurements over time, as well as the inclusion of calf as a random effect in the analyses examining the associations between TUS, CRS, and blood variables. The corresponding text and tables have been amended to reflect these updates. Detailed, point-by-point responses are provided below. We have attached a “clean” and “tracked changes” version of the document for your convenience.

L31 | "TUS scores (0: healthy to 3: severe consolidation)" – Incomplete scale; Table 2 shows scores up to 5. | Revise to: "TUS scores (0: healthy to 5: severe pneumonia)" and clarify in text that scores 4–5 were rare and collapsed into score 3 for analysis (L196–197). |

Now L31: We have revised this sentence to clarify values of TUS 4 and 5 were reclassified for the analysis.

L32 | "CRS (0: healthy to 3: severe clinical symptoms)" – Implies linear scale, but scoring is categorical. | Clarify: "CRS categories (0: healthy, 1: mild, 2: moderate, 3: severe)" to reflect ordinal nature.

Now L32 – L33: We have revised this sentence to your suggested wording.

L35 "3 or 7 days post-arrival (T3)" – Introduces variability without justification. | State: "T3 was defined as 3 or 7 days post-arrival due to logistical constraints; analysis adjusted for time."

We agree this is an issue, however we were not able to address it in the abstract due to the word limit. We have ensured it is mentioned clearly in the methods.

L41 "Total immunoglobulins and IgG were lower for TUS of 3" – IgG is part of total Ig; cannot be lower unless others decrease. | Recheck calculation: Total Ig = IgA + IgG + IgM. If IgG ↓ and total Ig ↓, clarify if IgA/IgM compensate.

Now L41: Thank you for your comment here. IgG was measured in g-L, whereas both IgA and IgM were measured in mg/L. Due to this different scale, IgA and IgM had minimal contributions to total IG, explaining why both IgG and Total IG decreased with higher TUS, regardless of changes in IgA and IgM.

L59 "Calves that are diseased pre-transport should not travel" – Assumes causality not tested. Temper: "Calves with clinical signs pre-transport may be at higher risk during transport."

Now L57 – L58: We have now reworded this sentence to:” Calves with clinical signs pre-transport may infect other calves and may be at higher risk of severe physiological decline and impaired welfare during transport [4]. “

L65 "CRS has low sensitivity (0.27–0.64)" – No citation for these values in transport context. | Cite source specifically for transport or pre-transport settings (e.g., Donlon et al. 2023, Ref 21).

Now L64: As far as I am aware, no study has tested the sensitivity of CRS in transport settings. Donlon et al. studied dairy heifers on farm. I have adjusted the text to reflect that these values represent on-farm assessments and not during transport.

L74 "TUS has sensitivities >0.85 and up to 0.94" – Cites [10,11], but [10] is subclinical, [11] lacks necropsy validation. Specify: "Sensitivities vary by population; in subclinical calves, Ollivett et al. [10] reported 0.85–0.94 vs. BAL, not necropsy."

Now L73 – L74: I think there might be some misunderstanding regarding these references. Ollivett (2015) assessed the sensitivities of both TUS and BALF, but compared them to subclinical cases of respiratory disease that were first assessed by clinical assessments and later verified by euthanasia and necropsies. They reported a sensitivity of 0.94 and specificity of 1.0 for TUS vs. Necropsies. Rabeling et al. (ref. 11) also euthanised case animals (diagnosed with more severe disease) and performed postmortem examinations of the lungs.

L89–94 Calves from farms and marts mixed without accounting for origin differences. Include "origin (farm vs. mart)" as a covariate in models or stratify analysis.

Now L88 – L94: This analysis was preliminary, with a very small sample size. When we attempted to include repeated measures and additional covariates, the models did not converge. Therefore, we treated the four time-points as independent observations, and we were mindful of this limitation in our discussion. We did include calf as random effect in the model to account for the calf effect as much as possible.

We did have information on source, cohort, age, weight, sex, and breed; however, including only a subset of these variables would have been inconsistent, so we opted for an “all or nothing” approach. For descriptive purposes, we presented changes over time, but these were not used in the statistical models. With a larger cohort, we would have included all covariates, but in this preliminary study, this was not feasible.

L95–98 Describes cohort differences (sex, age, breed) but no statistical adjustment. Adjust models for cohort (C1 vs C2) or include as random effect.

See previous comment.

L105 "Calves from livestock mart assessed on day of transport" – Timing differs from farm calves (L104: one day pre). Standardize or acknowledge bias in pre-transport assessment timing.

We have included an additional sentence in the limitations section of the discussion (L498 – 501) to mention the possible effect of sampling these calves on different days.

L122–124 All calves received antibiotics upon arrival – confounds disease progression. Acknowledge that metaphylaxis limits natural disease progression and affects TUS/CRS/blood interpretation.

Now L121 – 124: The antibiotic use was mentioned four times in various sections of the discussion. The further elaborate, I have included the following sentence in the limitations: ”Firstly, the batch antibiotic treatments in this study is likely to have limited disease progression, particularly limiting clinical signs of disease, though this is standard practice in veal systems, and therefore the presentation of the current data is still representative for calves transported long-distance to intensive veal systems. “

L130–133 TUS and CRS not performed on same day at T4/T5 (blood sampled 1–2 days apart). State: "Non-simultaneous sampling may reduce correlation; results interpreted with caution."

The antibiotic use was mentioned four times in various sections of the discussion. The further elaborate, I have included the following sentence in the limitations: ”Firstly, the batch antibiotic treatments in this study is likely to have limited disease progression, particularly limiting clinical signs of disease, though this is standard practice in veal systems, and therefore the presentation of the current data is still representative for calves transported long-distance to intensive veal systems. “

L134–136 No blood at T3, but TUS/CRS done – limits temporal comparison. | Justify gap and avoid claiming integrated assessment at T3.

Now L134 – 136: Thank you for your comment. In the text, I have clarified that T3 samples did not include blood samples due to logistical constraints. Even though this paper presents descriptive statistics in changes over time, the statistical analysis directly compared assessments methods, regardless of the time they were collected at. Therefore the lack of blood samples at T3 means this timepoint was omitted for TUS or CRS vs. Blood comparisons. There was no temporal assessments in the statistical analyses.

L147 "Depth of 11cm" – Not validated for calf thorax; may miss deep lesions. Cite justification or note potential for missed consolidations.

Ollivett (2018) reported a depth of 8 to 10 cm is appropriate for evaluating lungs in young cattle. The observer in this study was trained by J. Donlon who has used Ollivett’s system and is well practiced in using thoracic ultrasonography for determining lung disease in calves. The ultrasound machine used did not have a standard setting between 8 and 10 cm, thus 11cm was used to best identify lesions in the lungs of these unweaned animals.

L153 "6-tier TUS" – Inconsistent with Table 2 (6-point scale) and text (0–3 used). Correct: "A 6-point TUS scale was used, but scores 4–5 were combined with 3 due to low frequency."

Now L152 – 157: We have reworded this sentence to clarify the original observations used a 6-point scale.

L161–163 | Rectal temperature omitted at T1 and imputed at T4 – introduces bias. Exclude temperature from T1 CRS or use alternative scoring; do not impute without validation.

Now L162 – 163: Temperatures were never measured at T1, and therefore were not present at the CRS calculation for this timepoint. They were not imputed. For T4, as temperatures were recorded within 24h of CRS assessments, and due to the gradual disease pattern, we believed it was most reliable to still include these variables. This was however thoroughly included as a limitation in the discussion.

L167–168 CRS categories based on sum 0–1=0, 2–3=1, etc. – unequal bin

---

## [Editor Report · Decision Letter 1]

10 Nov 2025

Comparison of thoracic ultrasonography (TUS), clinical respiratory scoring (CRS), and blood analysis to evaluate respiratory dysfunction in transported calves

PONE-D-25-22875R1

Dear Dr. Van Dijk,

We’re pleased to inform you that your manuscript has been judged scientifically suitable for publication and will be formally accepted for publication once it meets all outstanding technical requirements.

Kind regards,

Mourad Mahmoud

Academic Editor

PLOS ONE
---

## [Editor Report · Acceptance letter]

PONE-D-25-22875R1

PLOS One

Dear Dr. van Dijk,

I'm pleased to inform you that your manuscript has been deemed suitable for publication in PLOS One. Congratulations! Your manuscript is now being handed over to our production team.

Kind regards,

on behalf of

Dr. Mourad Mahmoud

Academic Editor

PLOS One